# Non-pharmacological interventions for the reduction and maintenance of blood pressure in people with prehypertension: a systematic review protocol

Emma P Bray [ORCID],[1] Rachel F Georgiou,[1] Lucy Hives [ORCID],[2] Nafisa Iqbal,[1] Valerio Benedetto,[3] Joseph Spencer [ORCID],[4] Cath Harris,[3] Andrew Clegg,[3] Nefyn Williams [ORCID],[5] Paul Rutter,[6] Caroline Watkins[1]

For numbered affiliations see end of article.

**Correspondence to**
Dr Emma P Bray;
EBray@uclan.ac.uk

## ABSTRACT

**Introduction** Prehypertension is defined as blood pressure that is above the normal range but not high enough to be classed as hypertension. Prehypertension is a warning of development of hypertension as well as a risk for cardiovascular disease, heart attack and stroke. In the UK, non-pharmacological interventions are recommended for prehypertension management but no reviews have focused on the effectiveness of these types of interventions solely in people with prehypertension. Therefore, the proposed systematic review will assess the clinical effectiveness and cost-effectiveness of non-pharmacological interventions in reducing or maintaining blood pressure in prehypertensive people.

**Methods and analysis** This systematic review will follow the Preferred Reporting Items for Systematic Reviews and Meta-Analyses guidelines. The databases/trial registries that will be searched to identify relevant randomised controlled trials (RCTs) and economic evaluations include Medline, EMBASE, CINAHL, PsycINFO, CENTRAL, the WHO International Clinical Trials Registry Platform, ClinicalTrials.gov, Cochrane Library, Scopus and the International HTA Database. Search terms have been identified by the team including an information specialist. Three reviewers will be involved in the study selection process. Risk of bias will be evaluated using the Cochrane risk-of-bias tool for RCTs and the Consensus Health Economic Criteria list for economic evaluations. Findings from the included studies will be tabulated and synthesised narratively. Heterogeneity will be assessed through visual inspection of forest plots and the calculation of the $\chi^2$ and $I^2$ statistics and causes of heterogeneity will be assessed where sufficient data are available. If possible, we plan to investigate differential effects on specific subgroups and from different types of interventions using meta-regression. Where relevant, the Grading of Recommendations, Assessment, Development and Evaluations (GRADE) will be used to assess the certainty of the evidence found.

**Ethics and dissemination** Ethical approval is not needed. Results will be published in a peer-reviewed journal, disseminated via the wider study website and shared with the study sites and participants.

**Registration details** The review is registered with PROSPERO (CRD420232433047).

## STRENGTHS AND LIMITATIONS OF THIS STUDY

⇒ To our knowledge, the systematic review will be the first to provide an up-to-date synthesis on non-pharmacological interventions to reduce blood pressure in individuals in the prehypertensive range.

⇒ A comprehensive search strategy has been designed collaboratively with a skilled information specialist.

⇒ There will be no restrictions applied to the searches regarding language or date of publication.

⇒ This review will only include randomised controlled trials and comparative economic evaluations. Other study designs will be excluded.

## INTRODUCTION

People with blood pressure (BP) in the prehypertension (PHT) range (120–139/80–89 mm Hg) are at increased risk, compared with those with normal BP, of developing hypertension[1 2] and other cardiovascular disease (CVD)-related conditions, independent of progression to hypertension.[3–9] The estimated number of people with PHT is substantial, with around 40% of adults attending primary care clinics falling into this category.[10] BP exists on a continuum following a normal distribution, with risk of CVD increasing as BP increases. PHT, also known as 'high-normal', 'elevated' or 'raised' BP, lies between normal BP (<120/80 mm Hg) and hypertension (≥140/90 mm Hg).

PHT itself however is not regarded as an illness, rather it is a warning of 'an insidious progression' of BP to problematic levels.[11] It is starting to be recognised that prevention needs to be at the heart of future healthcare[12] with a change of emphasis from fixing ill health to protecting good health. PHT can be a useful sign to identify and alert those at risk of developing hypertension and CVD, so

they can take action to prevent, delay or reduce progression to disease status.[11]

In the UK, PHT does not require drug intervention like in hypertension. Rather, guidance[13] recommends that PHT management should be focused on lifestyle modification.

There is a plethora of primary research studies demonstrating that BP in people with PHT can be managed via various lifestyle interventions, including both single component interventions for example, exercise,[14–16] diet,[17 18] breathing exercises,[19 20] yoga,[21 22] etc, as well as multicomponent lifestyle interventions.[23 24]

Research has also demonstrated that lifestyle modifications at this 'pre-risk' stage could provide lifetime benefits; for example, by making lifestyle changes, people with PHT could significantly reduce both their risk of developing hypertension[25] and reverse PHT to normotensive levels.[26]

There are a number of published systematic reviews on this topic. However, these tend to focus on one type of non-pharmacological intervention such as exercise.[27] There are two existing reviews that assess the effectiveness of various non-pharmacological interventions on PHT.[28 29] However, both reviews included studies combining PHT and hypertensive participants making it impossible to determine what might be effective for those specifically with PHT. Additionally, neither included a cost-effectiveness review and the most recent review[28] only looked at outcomes at 4 weeks, which therefore does not inform us about the longer-term effectiveness of non-pharmacological interventions.

Therefore, an up-to-date review of current evidence on clinical effectiveness and cost-effectiveness of non-pharmacological interventions specifically for PHT would help guide research and practice in this area, with the potential to prevent not only progression to hypertension but also prevent CVD in the future. The aim of this review is to assess the clinical and cost-effectiveness of different non-pharmacological interventions for reducing BP and maintaining BP in adults with PHT.

## Review question

What is the clinical effectiveness and cost-effectiveness of non-pharmacological interventions for reducing and/or maintaining blood pressure in adults with prehypertension, compared with usual care, or other non-pharmacological interventions?

## METHODS AND ANALYSIS
### Study design

This systematic review will be conducted and reported following the Preferred Reporting Items for Systematic Reviews and Meta-Analyses (PRISMA) guidelines.

### Search strategy

Separate searches to identify randomised controlled trials (RCTs) or economic evaluations will be conducted.

Seven databases and trial registries will be searched to identify relevant RCTs: Medline, EMBASE, CINAHL, PsycINFO, Cochrane Central Register of Controlled Trials (CENTRAL), the WHO International Clinical Trials Registry Platform and ClinicalTrials.gov. Medline, EMBASE, CINAHL, PsycINFO, Cochrane Library via Wiley (all databases), Scopus and the International HTA Database will be searched to identify economic evaluations. We will screen the reference lists of all included studies and relevant review articles for any additional studies. There will be no restrictions applied to the searches regarding language or date of publication.

The search terms were identified by the team (including an information specialist) via an iterative process. The search strategies include keywords and subject headings relating to prehypertension which were combined with the Cochrane Highly Sensitive Search Strategy[30] for identifying Randomised Trials in Medline (sensitivity-maximising version) and the NHS EED Search Filter[31] to identify economic evaluations. The search strategies were adapted for use in each database and combined with database-specific search filters for RCTs and economic evaluations where these are available. The searches were conducted in May 2023. The search strategies for Medline can be found in online supplemental appendix 1.

### Study selection

Search results will be imported into EndNote (V.X9, Clarivate Analytics, Philadelphia, Pennsylvania), where duplicates will be removed. The final list for screening will then be imported into Rayyan. Screening for the studies of clinical effectiveness and cost-effectiveness will be done as two separate processes. One reviewer will screen all the titles and abstracts of the search results to identify relevant articles, and a second reviewer will review at least 10% to ensure consistency. A third reviewer will then check those studies deemed as potentially eligible for the next stage. One reviewer will then independently screen the full texts of the potentially eligible articles, and a second reviewer will check the ones selected for inclusion, to determine final inclusion in the review. Disagreements will be resolved via discussion and where necessary through the involvement of a third reviewer. The study selection process will be presented using a PRISMA flow diagram.

### Inclusion criteria

Only RCTs and comparative economic evaluations (including costs and/or consequences) will be eligible for inclusion in the review.

### Condition or domain being studied

Prehypertension

Studies that report data from samples with PHT AND hypertension (controlled or uncontrolled) will be included *only* if the data can be separated out so that only data regarding PHT can be extracted.

## Participants/population

Inclusion criteria: people aged 18 years old or older with blood pressure in the prehypertension range defined as systolic blood pressure 120–139 mm Hg and/or diastolic blood pressure 80–89 mm Hg.

Exclusion criteria: we will exclude data from participants that have/had a diagnosis of hypertension, are on antihypertensive medication, are pregnant, have a life-limiting disease or have previously had a cardiovascular event (eg, heart attack, stroke, transient ischaemic attack) or have a pre-existing cardiovascular disease.

## Intervention(s), exposure(s)

All non-pharmacological interventions will be considered, regardless of presentation or delivery mode (eg, face-to-face, group, one-to-one, self-directed, etc), for inclusion. This will include investigations of foods and/ or compounds found in foods (eg, caffeine, flavonoids).

## Comparator(s)/control

Studies will be included if they compare non-pharmacological interventions to usual care or if they compare different types of non-pharmacological interventions to each other.

## Context

Study setting may be primary or secondary care, or community settings.

## Main outcome(s)

Change in BP (both systolic and diastolic) between intervention and comparator will be the main outcome for the clinical effectiveness strand of our review and studies that report such change will be included. BP could be office, ambulatory or self-monitored readings. We will also include studies that report a change in cardiovascular morbidity (cardiovascular disease, hypertension, myocardial infarction or stroke/transient ischaemic attack).

For the cost-effectiveness strand, the main outcomes will be a change in costs, quality of life (QoL) and cost-effectiveness, and studies reporting these will be included.

All follow-up lengths will be included, with the differential effects associated with different durations explored in subgroup analyses.

## Measures of effect

The measures of effect for change in BP and change in QoL will be the mean difference, SD and 95% CI.

For occurrence of CVD, hypertension, stroke, transient ischaemic attack (TIA) and myocardial infarction (MI), the measure of effect will be the OR.

For cost-effectiveness, we will consider mean differences in costs and QoL between comparators (and any dispersion measures as reported by the included studies), as well as cost per marginal gain in the specific QoL measures employed by the studies (eg, quality-adjusted life years or QALYs).

## Data extraction (selection and coding)

Two reviewers will independently extract data from the agreed eligible studies. A coding scheme will be developed and piloted to ensure all relevant data are captured and to ensure both reviewers are extracting data in a standardised manner. It is envisaged that the following data will be extracted:

▶ *Study context:* authors, study title, journal, year of publication, location of study, funding details, conflicts of interest;

▶ *Design and methods:* method of randomisation, details of study sites (number, location, setting etc), variables of interest, data collection tools, planned follow-up timings, key attributes of economic evaluations (eg, perspective, design, time horizon, characteristics of cost and outcome data, cost-effectiveness measure employed);

▶ *Participants:* number, basic demographics (age, gender, ethnicity), blood pressure, comorbidities;

▶ *Interventions:* type of intervention, details of the intervention including length, frequency and duration, comparison, mode of delivery, provider, content and components;

▶ *Outcomes:* systolic blood pressure, diastolic blood pressure, presence of cardiovascular morbidity (hypertension, MI, stroke, TIA, cardiovascular disease), costs, QoL and cost-effectiveness at baseline and the stated follow-up time points of each study.

## Risk of bias (quality) assessment

Two reviewers will independently critically appraise the eligible studies. Risk of bias will be evaluated using the Cochrane risk-of-bias tool (RoB1) for RCTs and the Consensus Health Economic Criteria (CHEC) list for economic evaluations.[32] Risk of bias will be incorporated within the synthesis and interpretation of the evidence, using Grading of Recommendations, Assessment, Development and Evaluations (GRADE) framework.[33]

## Strategy for data synthesis

Studies will be synthesised through narrative reviews with tabulation of results of included studies. Where possible, treatment effects for all comparisons and outcomes will be synthesised through meta-analyses, with the approach taken depending on the outcome assessed and the data available. Dichotomous data will be presented as risk ratios with 95% CIs. Continuous data will be synthesised as weighted mean differences (MD) when outcomes are assessed on the same scale or standardised weighted mean difference (SMD) when different scales are used to measure the same underlying construct, with 95% CI. Where the outcomes represent time-to-event data (eg, overall survival), the (log) HR with 95% CI will be used as the summary measure. Heterogeneity will be assessed through visual inspection of forest plots and the calculation of the $\chi^2$ and $I^2$ statistics. Causes of heterogeneity will be assessed where sufficient data are available, including factors such as participant characteristics (eg, age, sex),

baseline BP, method of obtaining BP readings (office, ambulatory, home), duration of intervention, comorbidities and delivery/mode of intervention. Where appropriate, these will be investigated further through subgroup analyses and through meta-regression. Sensitivity analyses will explore possible causes of methodological heterogeneity, where sufficient data are available. This would include assessing the effects of studies that may be affected by factors such as risk of bias associated with allocation concealment, high loss to follow-up or lack of blinding in assessment of outcomes. Where data are missing, particularly measures of variation (eg, SD), we will contact study authors or impute values. It is likely that the analysis will focus on direct comparisons of intervention effects through pairwise meta-analyses. Where evidence allows, we will consider conducting network meta-analysis through both direct and indirect evidence within connected networks of trials. Pairwise meta-analyses of direct comparisons will be conducted using STATA V.17 (StataCorp, Texas, USA) or Comprehensive Meta-analysis V.4, while NMAs will be estimated using the WinBUGS software (V.1.4.3) (MRC Biostatistics Unit, Cambridge, UK) (http://www.mrc-bsu. cam.ac.uk/bugs/winbugs/contents.shtml).

## Analysis of subgroups or subsets

Depending on the variety of non-pharmacological interventions and studies' populations and on the available data, we plan to investigate differential effects on specific subgroups (eg, by age, sex, BP levels, comorbidities, health-related quality of life) and from different types of interventions (eg, by duration, setting, mode of delivery) using meta-regression. Where this quantitative approach is not possible, we will discuss the subgroup effects narratively.

## Assessment of certainty of evidence

Where relevant, we will assess the level of certainty of the evidence found in the included studies using the GRADE framework.[21]

## Patient and public involvement

Patients and the public were not directly involved in designing this systematic review protocol. The systematic review is connected to a larger programme of work called REVERSE (Risk Education InterVEntion for Raised blood preSsurE). REVERSE aims to investigate the feasibility of blood pressure self-monitoring for people with prehypertension, which was coproduced with members of the public with prehypertension and hypertension. The results of both the systematic review of non-pharmacological interventions and feasibility study of self-monitoring will help with coproducing future lifestyle interventions, with the aim of preventing hypertension and associated conditions, which will be tested in future effectiveness studies.

## ETHICS AND DISSEMINATION

Ethical approval is not needed for this systematic review. The review is registered with PROSPERO (CRD420232433047) and any important protocol amendments will be updated on the record. It is intended to publish the completed review in a peer-reviewed journal. Results will also be disseminated via the REVERSE study website and will be shared with the REVERSE study sites and participants.

**Author affiliations**
[1]Stroke Research Team, University of Central Lancashire, Preston, UK
[2]School of Community Health and Midwifery, University of Central Lancashire, Preston, UK
[3]Synthesis, Economic Evaluation and Decision Science Group, University of Central Lancashire, Preston, UK
[4]Research Facilitation and Delivery Unit, University of Central Lancashire, Preston, UK
[5]Primary Care, Department of Primary Care and Mental Health, University of Liverpool, Liverpool, UK
[6]Pharmacy Practice, School of Biomedical Sciences, University of Portsmouth, Portsmouth, UK

**Contributors** EPB was responsible for the conception, design and writing of the manuscript. LH, JS, EPB, RFG, CW, VB, NW, PR and AC contributed to the conception and design. EPB, LH, VB and NI contributed to the writing of the manuscript. CH designed the search strategy and performed the preliminary database searches. All authors read and approved the final manuscript. EPB is the guarantor of the review.

**Funding** EPB holds a NIHR RfPB grant (NIHR201028). This work is related to this funding but is not directly funded by it. The views expressed are those of the author(s) and not necessarily those of the NIHR or the Department of Health and Social Care.

**Competing interests** VB, CH, AC, JS and CW are partly funded by the National Institute for Health Research Applied Research Collaboration North West Coast (NIHR ARC NWC). The views expressed are those of the authors and not necessarily those of the NHS, the NIHR or the Department of Health and Social Care.

**Patient and public involvement** Patients and/or the public were not involved in the design, or conduct, or reporting, or dissemination plans of this research.

**Patient consent for publication** Not applicable.

**Ethics approval** Not applicable.

**Provenance and peer review** Not commissioned; externally peer reviewed.

**Data availability statement** No data are available.

**ORCID iDs**
Emma P Bray http://orcid.org/0000-0001-9882-3539
Lucy Hives http://orcid.org/0000-0003-4125-4034
Joseph Spencer http://orcid.org/0000-0003-3723-7629
Nefyn Williams http://orcid.org/0000-0002-8078-409X

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
