## [Reviewer comments · BMJ Open]

ARTICLE DETAILS

TITLE (PROVISIONAL)	Non-pharmacological interventions for the reduction and maintenance of blood pressure in people with prehypertension: A systematic review protocol
AUTHORS	Bray, Emma; Georgiou, Rachel; Hives, Lucy; Iqbal, Nafisa; Benedetto, Valerio; Spencer, Joseph; Harris, Cath; Clegg, Andrew; Williams, Nefyn; Rutter, Paul; Watkins, Caroline

VERSION 1 – REVIEW

REVIEWER	Cobucci, Ricardo Ney Universidade Federal do Rio Grande do Norte, Health Woman Graduate Program
REVIEW RETURNED	24-Aug-2023

GENERAL COMMENTS	The authors present a systematic review protocol with the objective of assessing the clinical- and cost-effectiveness of nonpharmacological interventions in reducing or maintaining blood pressure in pre-hypertensive people. It is well written and covers most of the PRISMA-P recommendations. I only suggest minor corrections to improve the manuscript: Abstract: include in Methods and analysis that GRADE will be used to assess the certainty of the evidence. Introduction: add a paragraph placing updated references with the main non-pharmacological measures used in adults with PHT and if there are gaps to be answered about their effectiveness and cost-effectiveness in this population.(https://doi.org/10.3389/fpubh.2022.1051581/ https://doi.org/10.1186/s12875-022-01884-8). In addition, add what this systematic review will add to the results of this one published and available at https://doi.org/10.3389/fpubh.2022.1051581. METHODS AND ANALYSIS: correct the phrase "Studies will also be included where samples contain people both with PHT AND hypertension or normotension if the data for the PHT group is presented and can be extracted separately" in lines 12, 13 and 14, as it is not consistent with the other sentence "We will exclude studies where participants have/had a diagnosis of hypertension, are on anti-hypertensive medication" in lines 24 and 25. Finally, I suggest the addition of the discussion section with a maximum of 3 paragraphs in which the authors will place the expected results, as a systematic review evaluating efficacy and cost-effectiveness of non-pharmacological measures can impact clinical decision-making and cost reduction in health and likely limitations of the study, such as heterogeneity, scarcity of economic evaluation studies, and high risk of bias and low certainty of evidence.
--

REVIEWER	Sun, Dianjun Center for Endemic Disease Control, Chinese Center for Disease Control and Prevention, Harbin Medical University
REVIEW RETURNED	01-Sep-2023

GENERAL COMMENTS	This review aims to assess the clinical and cost-effectiveness of different non-pharmacological interventions for reducing BP and maintaining BP in adults with prehypertension. The study process is well-designed and clearly described. However, I did not see the main data outcome except for some general statements, the bias analysis or heterogeneity assessment methods were not shown clearly, so please improve and refine these expected protocols.
--

VERSION 1 – AUTHOR RESPONSE

Reviewer 1	
Abstract: include in Methods and analysis that GRADE will be used to assess the certainty of the evidence.	The following text has been added to p2 “Where relevant, the Grading of Recommendations, Assessment, Development, and Evaluations (GRADE) will be used to assess the certainty of the evidence found.”
Introduction: add a paragraph placing updated references with the main non-pharmacological measures used in adults with PHT and if there are gaps to be answered about their effectiveness and cost-effectiveness in this population. (https://doi.org/10.3389/fpubh	We have added a few sentences on page 2/3 summarising the studies available that have looked at various non-pharmacological interventions in PHT.
In addition, add what this systematic review will add to the results of this one published and available at https://doi.org/10.3389/fpubh.2022.1051581 .	Thank you for bringing this reference to our attention. We looked at the studies included in the review by Shao and we feel that our review addresses a number of gaps and issues raised by it and the Fu review:  1. Both reviews include studies that have data combined from people pre-hypertension and stage 1 hypertension/hypertension. Ours will focus solely on PHT 2. The follow-up period in the Shao review was up to 4-weeks. Ours will include much wider follow-up periods allowing us to look at which interventions are effective longer-term. 3. Our review will include both a clinical and cost-effectiveness analysis which neither of the published reviews include. 4. Our review will also look at additional outcomes (morbidity, QoL) giving a broader picture of the effectiveness of the interventions. 5. It will also provide an update of the literature. We have added some text around this on p3
METHODS AND ANALYSIS: correct the phrase "Studies will also be included where	We have altered the wording to make this clearer. On p4 under the heading Condition/

samples contain people both with PHT AND hypertension or normotension if the data for the PHT group is presented and can be extracted separately" in lines 12, 13 and 14, as it is not consistent with the other sentence "We will exclude studies where participants have/had a diagnosis of hypertension, are on anti-hypertensive medication" in lines 24 and 25.	domain being studied the text now reads "Studies that report data from samples with PHT AND hypertension (controlled or uncontrolled) will be included only if the data can be separated out, so that only data regarding PHT can be extracted."
Finally, I suggest the addition of the discussion section with a maximum of 3 paragraphs in which the authors will place the expected results, as a systematic review evaluating efficacy and cost-effectiveness of non-pharmacological measures can impact clinical decision-making and cost reduction in health and likely limitations of the study, such as heterogeneity, scarcity of economic evaluation studies, and high risk of bias and low certainty of evidence.	Thank you for your suggestion. The guidance for submission of protocols does not include a discussion section https://bmjopen.bmj.com/pages/authors/#protocol-hence-why-we-did-not-include-one. Although in principle we are happy to include a discussion section, it would purely be speculative, as at this point we do not know what the findings will be regarding the issues the reviewer suggests e.g., we do not know if there will be a high risk of bias etc. Therefore, we do not feel this would add anything scientific or concrete to this review.
Reviewer 2	
This review aims to assess the clinical and cost-effectiveness of different non-pharmacological interventions for reducing BP and maintaining BP in adults with prehypertension. The study process is well-designed and clearly described.	Thank you
However, I did not see the main data outcome except for some general statements, the bias analysis or heterogeneity assessment methods were not shown clearly, so please improve and refine these expected protocols.	Thank you for these comments, and we recognise the need to better signpost these elements within our protocol:  • We have modified the section on 'Main outcome(s)' on p. 5 by clearly identifying which main outcomes characterise the clinical effectiveness (change in BP) and the cost-effectiveness strands (change in costs, quality of life (QoL) and cost-effectiveness) of our review. If useful, we also note that examples of specific measures are also provided within the sub-section on 'Measures of effect' on p. 5. • We have added a sentence within the section 'Risk of bias (quality) assessment' on p. 5 about incorporating risk of bias into the synthesis and interpretation of the results using GRADE. • On the heterogeneity assessment methods, we feel that we included the related details within the section 'Strategy for data synthesis' on p. 6 (and associated sub-sections on 'Analysis of subgroups or subsets' and 'Assessment of certainty of evidence'). However, if the reviewer thinks that there are specific aspects that are currently missing, then we would be happy to take them into consideration and potentially add them onto that section.

VERSION 2 – REVIEW

REVIEWER	Cobucci, Ricardo Ney Universidade Federal do Rio Grande do Norte, Health Woman Graduate Program
REVIEW RETURNED	14-Oct-2023

GENERAL COMMENTS	The authors have satisfactorily met most of the reviewers' recommendations and the revised manuscript can be accepted. Congratulations.
---

REVIEWER	Sun, Dianjun Center for Endemic Disease Control, Chinese Center for Disease Control and Prevention, Harbin Medical University
REVIEW RETURNED	15-Dec-2023

GENERAL COMMENTS	No
----

VERSION 2 – AUTHOR RESPONSE